# Design of a Convolutional Neural Network Accelerator Based on On-Chip Data Reordering

**Yang Liu** [1,2]**, Yiheng Zhang** [1]**, Xiaoran Hao** [1,]*****, Lan Chen** [1,]*****, Mao Ni** [1]**, Ming Chen** [1] **and Rong Chen** [1]

[1]  Institute of Microelectronics of the Chinese Academy of Sciences, Beijing 100029, China; liuyang2022@ime.ac.cn (Y.L.); zhangyiheng1@ime.ac.cn (Y.Z.); nimao@ime.ac.cn (M.N.); chenming@ime.ac.cn (M.C.); chenrong@ime.ac.cn (R.C.)
[2]  University of Chinese Academy of Sciences, Beijing 100049, China
*   Correspondence: haoxiaoran@ime.ac.cn (X.H.); chenlan@ime.ac.cn (L.C.)

**Abstract:** Convolutional neural networks have been widely applied in the field of computer vision. In convolutional neural networks, convolution operations account for more than 90% of the total computational workload. The current mainstream approach to achieving high energy-efficient convolution operations is through dedicated hardware accelerators. Convolution operations involve a significant amount of weights and input feature data. Due to limited on-chip cache space in accelerators, there is a significant amount of off-chip DRAM memory access involved in the computation process. The latency of DRAM access is 20 times higher than that of SRAM, and the energy consumption of DRAM access is 100 times higher than that of multiply–accumulate (MAC) units. It is evident that the "memory wall" and "power wall" issues in neural network computation remain challenging. This paper presents the design of a hardware accelerator for convolutional neural networks. It employs a dataflow optimization strategy based on on-chip data reordering. This strategy improves on-chip data utilization and reduces the frequency of data exchanges between on-chip cache and off-chip DRAM. The experimental results indicate that compared to the accelerator without this strategy, it can reduce data exchange frequency by up to 82.9%.

**Keywords:** hardware accelerator; convolutional neural networks (CNN); data reuse





## 1. Introduction

Convolutional neural networks (CNNs) have achieved outstanding predictive accuracy in a wide range of computer vision tasks, including image recognition [1–5], object tracking [6,7], scene labeling [8,9], and detection [10–14], in recent years. Nevertheless, the computation involved in convolutional neural networks necessitates billions of multiply–accumulate operations and millions of data points (comprising weight data and feature map data). The huge amount of computation and data means a huge cost of hardware resources. How to utilize limited hardware resources to complete efficient neural network calculations is the key to accelerator design. In the process of convolutional neural network computation, the delay and power consumption of data transfer and memory access far exceed that of logical computation [15,16]. Therefore, reducing unnecessary data transfer and memory access without affecting normal calculations has become the key to improving accelerator energy efficiency.

In recent years, many groups have proposed their architectural design ideas for the design of CNN accelerators. The accelerator architecture proposed in [17] consists of a processing engine (PE) array and a multi-level memory structure. Multi-level memory structures include off chip DRAM, global buffer, on-chip network (NOC), and registers in PE. In [7,18], a deep convolutional neural network (DCNN) acceleration architecture called deep neural architecture (DNA) is proposed, providing reconfigurable computing modes for different models. For different convolutional layers, DNA can reconfigure their data paths to support mixed data reuse patterns. For different convolutional parameters,

DNA can reconfigure its computing resources to support highly scalable convolutional mapping methods. The above CNN accelerator architectures are different, but they all include PE arrays and various levels of memory for storing data. This is also the basic design architecture followed by our designed accelerator.

In CNNs, convolutional computations constitute over 90% of the overall computations [19,20]. Therefore, accelerating convolutional computation is the key to improving the computational speed of convolutional neural networks. Convolutional operations have the characteristic of high data repetition between adjacent computations, which can be utilized to achieve data reuse and improve computational efficiency. For this purpose, researchers have proposed various data reuse schemes [21], including weight stationary (WS) dataflow [17], input stationary (IS) dataflow, output stationary (OS) dataflow, and no local reuse (NLR) dataflow. The WS/IS dataflow keeps the weight data/input feature map data stationary on PE and reuses the weight data/input feature map data. Partial sums (psums) flow and accumulate through neighboring PEs to obtain the final result. In the output stationary (OS) dataflow, partial sums remain stationary while weight data and feature map data are moved. The partial sums are continuously accumulated at their original locations, resulting in the final computation result. The above three methods are all to keep some data stationary, increase the reuse rate of this part of the data, but will increase the number of data movements for the remaining parts. The no local reuse (NLR) dataflow does not temporarily store any data on the PE. Instead, it utilizes the saved space to increase the size of the global buffer, allowing it to store more data. This reduces the frequency of data exchanges with the DRAM. However, it increases the number of data exchanges between the global buffer and the PE, which typically results in higher power consumption compared to the power consumed by data movement between PEs. In [21], a dataflow called row stationary (RS) is introduced, where feature maps and weight data are input to the PE in the form of row vectors. Each PE can independently perform a 1-D convolution primitive and store partial sums in the spads within the PE. Partial sums can be transmitted between neighboring PEs through the network on chip (NOC), which increases data reuse and reduces the number of DRAM access operations. However, because a single PE needs to complete complex tasks such as data storage, 1-D convolution operations, and data transmission, the design of PE is relatively complex. The dataflow in this article adopts the OS dataflow, which can greatly reduce the movement of partial sums and simplify the PE function. Meanwhile, we design a data reuse module that improves the reuse rate of feature maps and weight data, thereby reducing the data exchange with off-chip DRAM.

In this article, we designed a universal and configurable CNN hardware accelerator. We have fully absorbed the experience of previous designs and focused on three main parts in accelerator architecture design: memory, computing, and control. To optimize data reuse, we adopted the OS dataflow proposed in [17]. We also designed the Data_reuse module, which enabled on-chip data reordering based on hardware. This optimization improved the dataflow, reduced the number of on-chip and off-chip communications, and further enhanced the efficiency of the accelerator.

The main contributions of this article are as follows:

- We implemented a CNN accelerator and proposed a PE array structure with a higher utilization rate of PE resources compared to the systolic array [22];
- We designed and implemented a data reuse module to improve the utilization of on-chip data that can significantly reduce the number of communications between on-chip and off-chip memory;

The remaining chapters of the article are organized as follows. Section 2 introduces the overall architecture design of the accelerator. Section 3 describes the design of specific modules in the accelerator. Section 4 discusses the key factors influencing the accelerator design. Section 5 presents the design of the Data_reuse module. Section 6 provides an analysis of the experimental results. The conclusions of the article are presented in Section 7.

## 2. The Accelerator Architecture

The overall structure of the CNN accelerator is shown in Figure 1, which consists of three main components: memory, PE array, and control logic. The memory is used to store input feature maps, weights, and output feature map data. The PE array is tasked with executing the multiplication and accumulation computations involved in convolution operations. The control logic is responsible for aligning and importing weight data and feature map data inside the chip into the PE array. It is also responsible for exporting the computed results from the PEs as output.

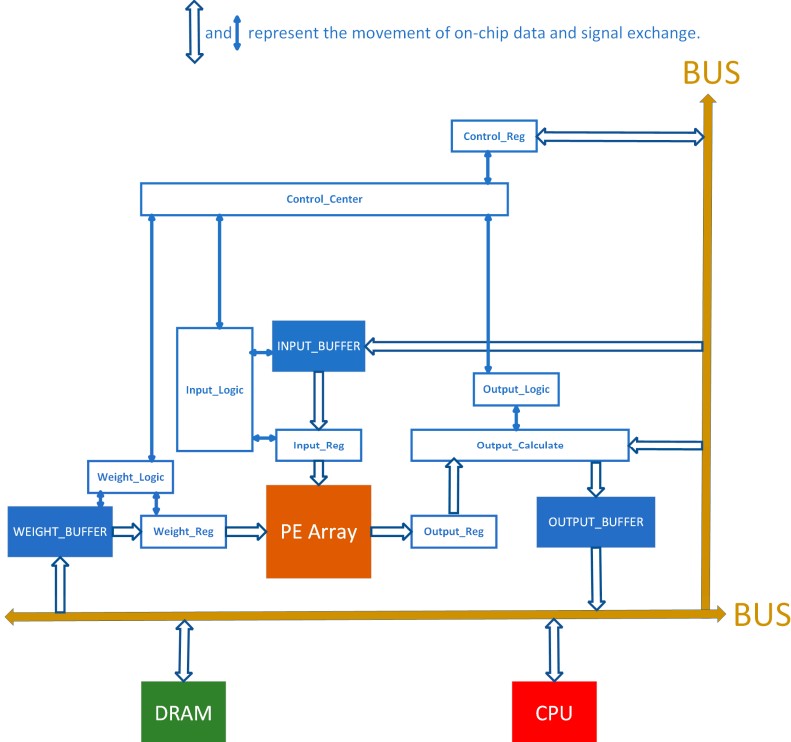

**Figure 1.** The overall architecture of the CNN accelerator.

The relationship between the three components mentioned above is illustrated in Figure 2. The control logic first reads off-chip data into the on-chip memory. Then, the data are read from the on-chip memory into the PE array for computation. The computed results are output to the on-chip memory for temporary storage and eventually transferred to off-chip memory.

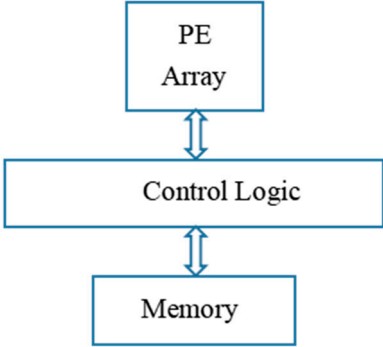

**Figure 2.** Relationship between modules of the CNN accelerator.

## 3. Partial Module Design

### 3.1. PE Array

The PE array design is shown in Figure 3, and the array size is configurable. We chose this array structure because compared to the systolic array (Figure 4b), our designed array (Figure 4a) can enter full load operation immediately after data transmission starts, while the systolic array needs to wait for several clocks before entering full load operation. Our array has higher parallelism. The size is determined by the size of the input feature map and the structure of the neural network. In this design, the PE array size is 32 rows and 26 columns. The data input method in the array is such that different weight data are input in different rows, and different tiles of the same feature map data are input in different columns. In each computation round, 26 different data points from each of the 32 different output feature maps are obtained, resulting in a total of 832 output data points. At 100 MHz, the throughput can reach 166.4 GOPS.

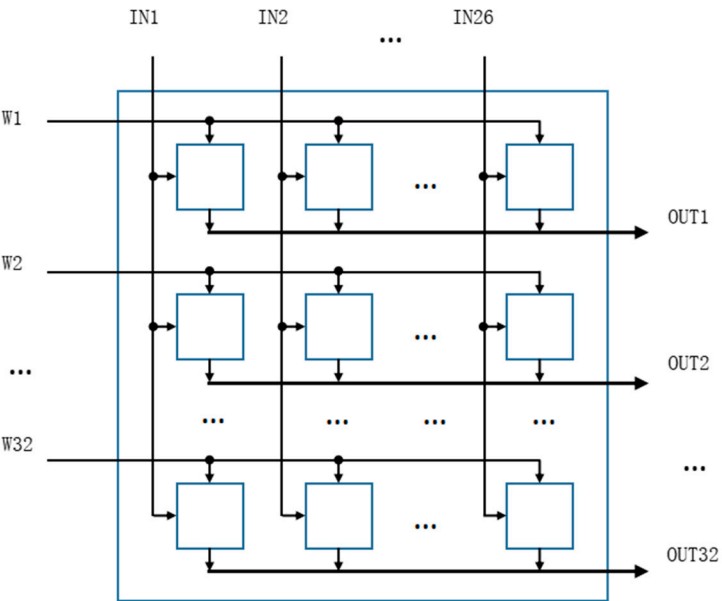

**Figure 3.** PE Array.

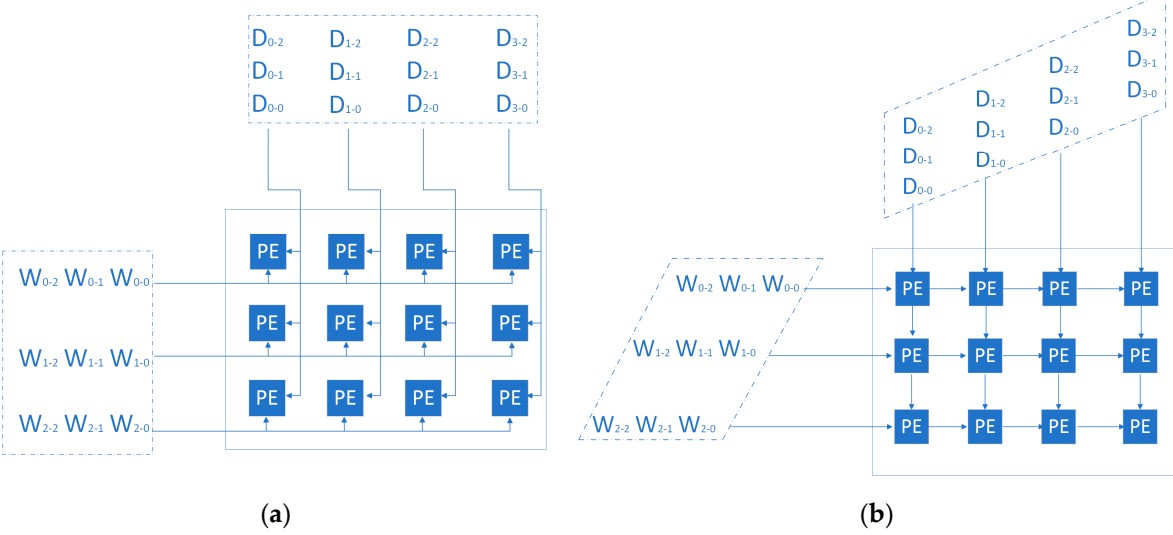

**Figure 4.** (**a**) The array we designed. (**b**) systolic array.

The structure of each PE unit in the PE array is shown in Figure 5. It mainly consists of a multiplier, accumulator, and register. The input data S (16, x) is a 16-bit signed number, where the fractional part width is x. Since the fractional part width varies for each layer, the variable x is used here to represent it. The output result of the signed multiplier has a width of 32 bits. To ensure that the addition result does not overflow, the accumulator is designed with a width of 45 bits. Finally, the width is corrected back to 16 bits by truncation.

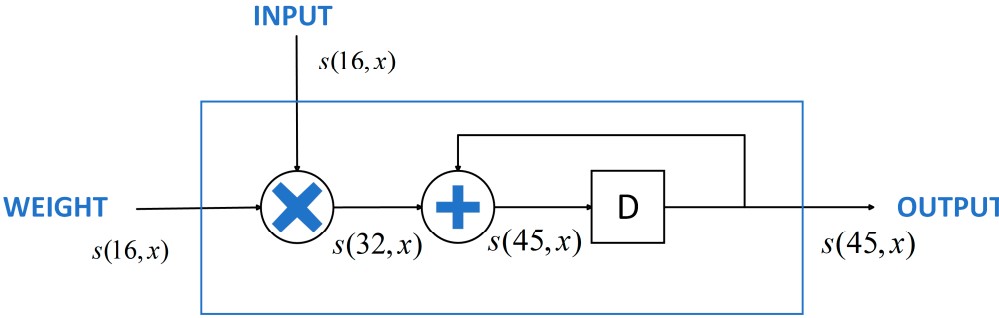

**Figure 5.** Single PE Structure.

### 3.2. BUS Interface Module

The bus interface module is shown in Figure 6. This module is mainly responsible for the data exchange between the on-chip memory (WBuf, InBuf, OutBuf, BiasBuf) and the off-chip bus. This module is also responsible for receiving control signals from the CPU and storing them in on-chip functional registers. These registers provide corresponding control signals to the control module.

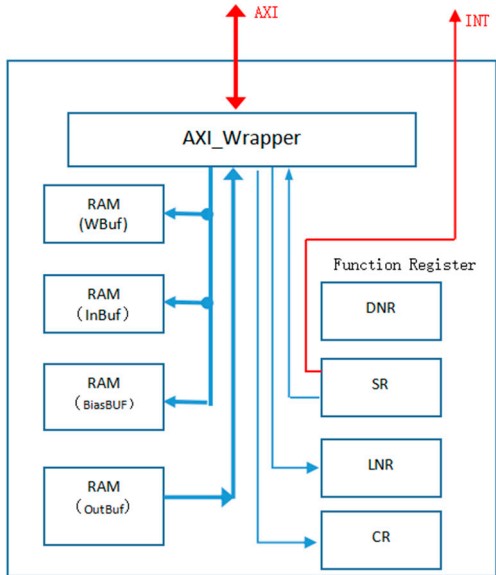

**Figure 6.** BUS Interface Module.

### 3.3. Cache and Register Access Control Module

The cache and register access control module, as shown in Figure 7, is responsible for issuing address signals to various on-chip memory spaces, receiving the corresponding data stored at the addresses, and then sending it to the computation units or off-chip memory. Moreover, it oversees the scheduling of control signals stored in diverse control registers, efficiently managing the operations of the remaining modules.

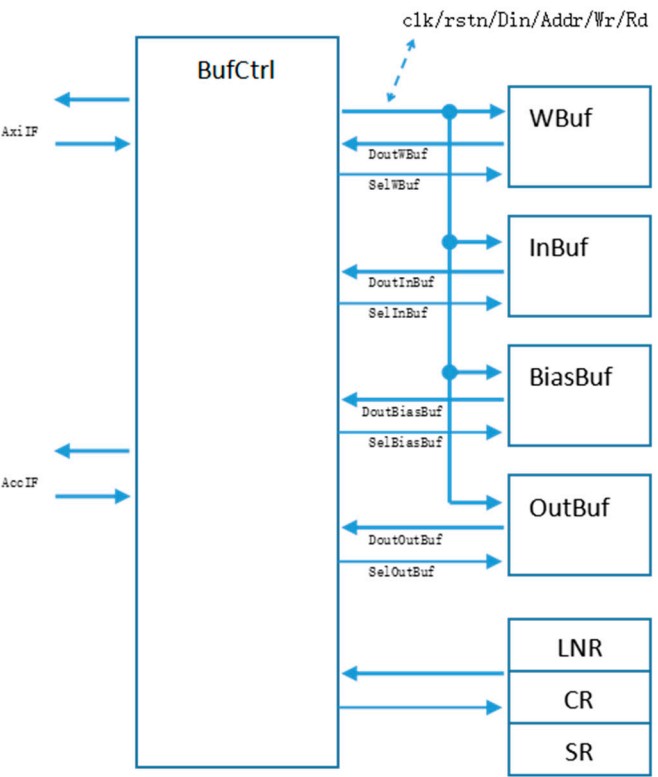

**Figure 7.** Cache and Register Access Control Module.

## 4. Data Reusability Analysis

The convolution operation process is shown in Figure 8. The essence of convolution is to multiply the corresponding position data between the input feature map and the weights and then accumulate them [23]. For the input feature map, there is data redundancy between adjacent convolution tiles. If data reuse is not considered, directly inputting the data required for each convolution calculation in sequence would result in multiple off-chip memory accesses for the same data, leading to a decrease in the utilization of on-chip memory. Figure 9 shows the data reusability analysis of convolutional operations. Taking a single-channel input feature map of size $13 \times 13$ as an example, assuming a convolution kernel size of $3 \times 3$ and a stride of 1, the gray area represents the zero-padding region. The numbers on each data element represent the number of times each element is used after completing the convolution operation on the entire feature map. If only the data involved in the computation is arranged in the calculation order and input to the PE array for computation, it results in a high degree of redundant off-chip data reading for the original data (excluding the data in the gray area). The number of redundant readings for each element, as shown in Figure 8, increases for elements closer to the center of the image. The maximum number of redundant readings is 9, while the minimum is 4. Taking into account the number of redundant readings for the gray area data (zero-padding data), it can be considered that each original datum is read redundantly 9 times. This is a huge number, which is equivalent to expanding the original data volume to 9 times the previous one. This results in a significant increase in the number of data exchanges between on-chip memory and off chip DRAM, leading to a decrease in accelerator speed and an increase in power consumption.

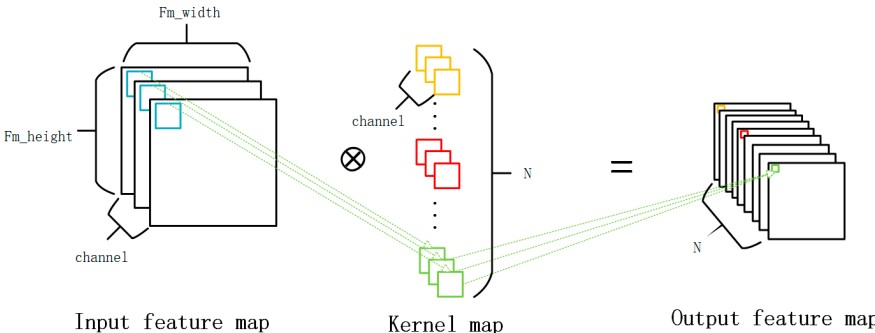

**Figure 8.** The convolution calculation process, where N represents the number of convolution kernels.

| 1 | 2 | 3 | 3 | 3 | 3 | 3 | 3 | 3 | 3 | 3 | 3 | 3 | 2 | 1 |
|---|---|---|---|---|---|---|---|---|---|---|---|---|---|---|
| 2 | 4 | 6 | 6 | 6 | 6 | 6 | 6 | 6 | 6 | 6 | 6 | 6 | 4 | 2 |
| 3 | 6 | 9 | 9 | 9 | 9 | 9 | 9 | 9 | 9 | 9 | 9 | 9 | 6 | 3 |
| 3 | 6 | 9 | 9 | 9 | 9 | 9 | 9 | 9 | 9 | 9 | 9 | 9 | 6 | 3 |
| 3 | 6 | 9 | 9 | 9 | 9 | 9 | 9 | 9 | 9 | 9 | 9 | 9 | 6 | 3 |
| 3 | 6 | 9 | 9 | 9 | 9 | 9 | 9 | 9 | 9 | 9 | 9 | 9 | 6 | 3 |
| 3 | 6 | 9 | 9 | 9 | 9 | 9 | 9 | 9 | 9 | 9 | 9 | 9 | 6 | 3 |
| 3 | 6 | 9 | 9 | 9 | 9 | 9 | 9 | 9 | 9 | 9 | 9 | 9 | 6 | 3 |
| 3 | 6 | 9 | 9 | 9 | 9 | 9 | 9 | 9 | 9 | 9 | 9 | 9 | 6 | 3 |
| 3 | 6 | 9 | 9 | 9 | 9 | 9 | 9 | 9 | 9 | 9 | 9 | 9 | 6 | 3 |
| 3 | 6 | 9 | 9 | 9 | 9 | 9 | 9 | 9 | 9 | 9 | 9 | 9 | 6 | 3 |
| 3 | 6 | 9 | 9 | 9 | 9 | 9 | 9 | 9 | 9 | 9 | 9 | 9 | 6 | 3 |
| 3 | 6 | 9 | 9 | 9 | 9 | 9 | 9 | 9 | 9 | 9 | 9 | 9 | 6 | 3 |
| 2 | 4 | 6 | 6 | 6 | 6 | 6 | 6 | 6 | 6 | 6 | 6 | 6 | 4 | 2 |
| 1 | 2 | 3 | 3 | 3 | 3 | 3 | 3 | 3 | 3 | 3 | 3 | 3 | 2 | 1 |

**Figure 9.** Analysis of Repetitiveness in Convolutional Operations. The numbers represent the number of times each data is reused. Different colors represent different categories. Gray represents zero-padded data. White represents data from the original image corners. Blue represents data from the original image edges. Green represents data from the original image interior.

There are two approaches to storing and reading feature map data. The first approach involves arranging the feature map data to be computed in the desired calculation order using a software algorithm (which may result in a large amount of redundant data). These arranged data are then stored off-chip and subsequently input into the on-chip memory. Finally, the data are fed into the PE array for computation. This method, as shown in Figure 10, results in a significant amount of redundancy in the data involved in adjacent computations. As a result, the effective input data are reduced, leading to a significant increase in the number of data transfers from off-chip to on-chip.

Another approach is to input the required original feature map data (without sorting and without any redundant data) into the on-chip cache. Then, through the on-chip data reuse module, the data are transformed into a stream that is arranged in the desired calculation order and fed into the PE array for computation. Compared to the first approach, this method utilizes on-chip hardware modules to perform data reordering, significantly reducing the number of data transfers from off-chip to on-chip. As a result, it improves the computational efficiency and energy efficiency of the system. The specific circuit implementation of the second approach is presented in Section 5.

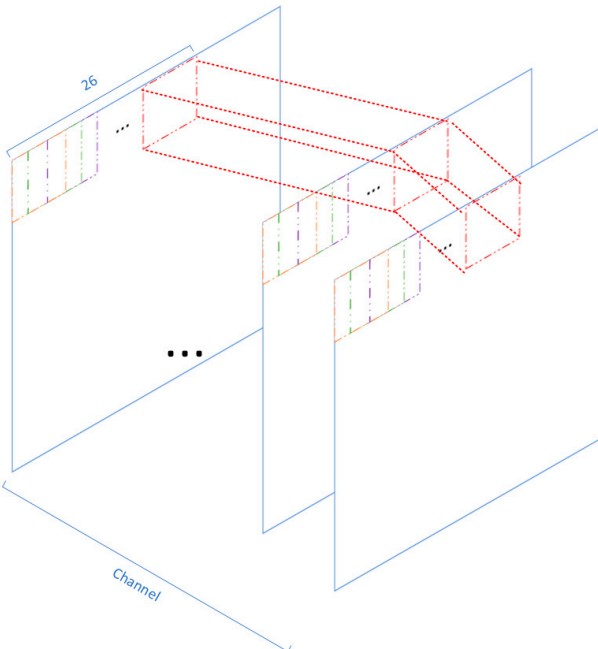

**Figure 10.** Original feature map storage method. The numbers represent the quantity of data tiles. Blue represents the total image data. Orange, green, and purple represent different data tiles. Red represents a complete three-dimensional tile of data.

From the characteristics of convolutional operations, it can be observed that the reuse rate of the weight data is also very high. There are two methods for storing and accessing weight data. The first method involves sorting the weight data in advance according to the order of convolution calculations using software algorithms, storing them off-chip, reading them from off-chip DRAM into on-chip caches, and finally, inputting them into the PE array for computation. This method simplifies the design of the weight control module but results in a significant amount of redundant weight storage and a high number of data movements between off-chip and on-chip caches. The number of times each weight datum is redundantly read equals the number of elements in a single-channel output feature map. The second method, which is a proposed method, involves storing non-redundant weight data in on-chip caches, and the weight control module controls the readout of cache data. This allows the computation process that requires the same weight data to repeatedly read the cache data until the weight data are no longer needed. The second method ensures that the weight data are replaced only when they are fully utilized, greatly reducing the number of data exchanges between on-chip caches and off-chip weight data. The comparison of the two methods is shown in Table 1.The specific design of the weight control module for the second method is provided in Section 5.

**Table 1.** Comparison between the first method and the second method.

|  | The First Method | The Second Method |
| --- | --- | --- |
| Do complex software sorting algorithms need to be used? | Yes | No |
| Are there duplicate data in on-chip memory space? | Yes | No |
| Does a complete convolution operation process require exchanging weight data with off-chip memory during the process? | Yes | No |

## 5. Hardware Circuit Implementation of Data Reordering

### 5.1. Feature Map Data Reuse Module

The fundamental reason for the reuse of feature map data is determined by the characteristics of convolutional operations. As shown in Figure 11, there are four tiles involved in the convolution operation in the X and Y directions. Due to their adjacent positions, there is data overlap between these tiles. If we do not consider reusing these overlapping data, it would result in a significant increase in data volume. As shown in Figure 11, the original data size is $4 \times 5$, totaling 20, while the actual amount of data input from off-chip is $3 \times 3 \times 4$, totaling 36. There are 16 data entries that have been duplicated.

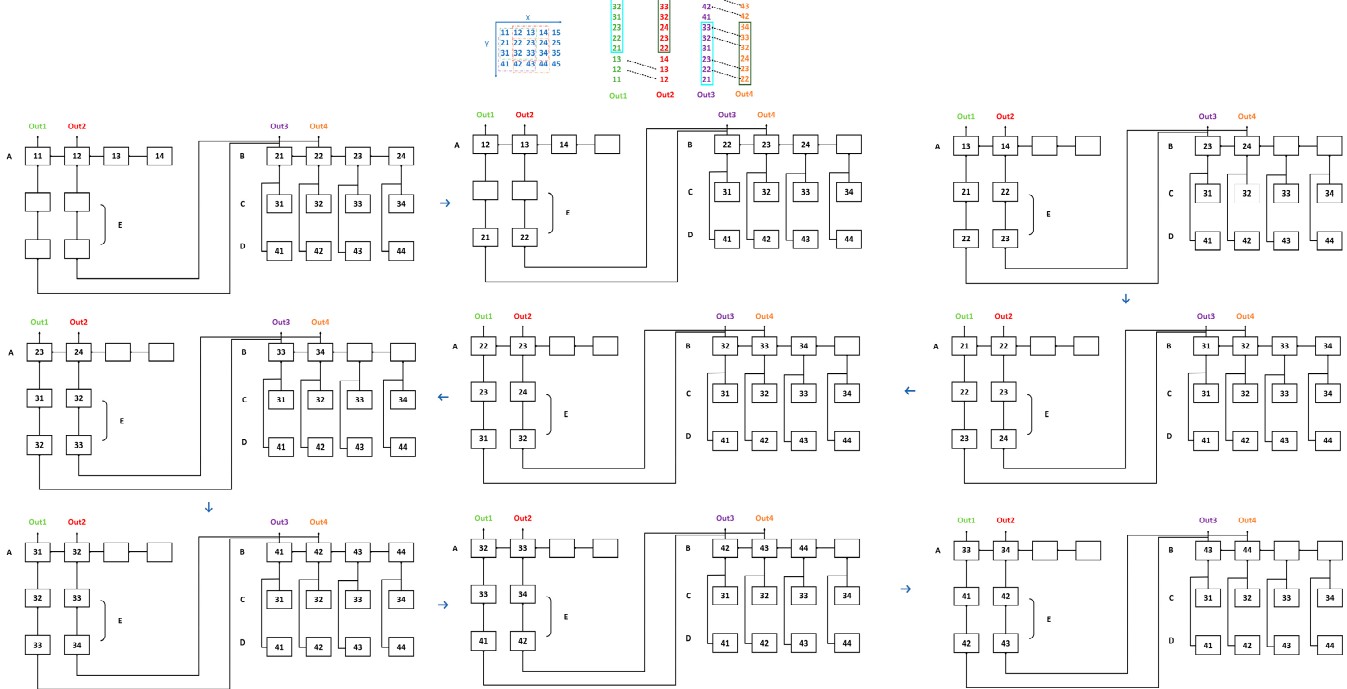

**Figure 11.** Data_reuse module and the dataflows in the Data_reuse module. The letters A, B, C, D, and E represent different types of registers. Registers A, B, and E are responsible for data movement, while registers C and D are responsible for data storage.

To address this issue, we propose a configurable hardware circuit structure that improves data reuse. As shown in Figure 11, we assume that the input feature map has 1 channel, the data size to be processed is $4 \times 5$, the convolution kernel size is $3 \times 3$, and the stride is 1. Taking the example of four output ports, Out1, Out2, Out3, and Out4, each port outputs the tiled data corresponding to the respective color. The original data only need to be input once. Data reuse in the X direction is achieved through data shifting between two sets of registers, A and B. Data reuse in the Y direction is achieved by delaying the input using register E. The C and D sets of registers serve as temporary storage for data, waiting to transfer the data to the B set of registers. This example demonstrates that there are 2 convolution operation tiles in the X direction and 2 convolution operation tiles in the Y direction. The flow of data in the Data_reuse module is shown in Figure 11. The number of tiles determines the number of sets of registers in the circuit, namely, A, B, C, D, and E. $RN_X$ represents the number of registers per group for A, B, C, and D. It can be calculated using Equation (1), where $KN$ represents the width of the convolution kernel, and $N_X$ represents the number of tiles in the X direction. The number of registers in the E group is determined by Equation (2), where $RN_E$ represents the number of registers per group for the E group. The term "per group" is used because there may not be only one group of E registers. It is determined by the number of tiles in the Y direction, denoted

as $N_Y$. $EN_Y$ represents the number of groups of E registers, and its size is determined by Equation (3). The E registers serve as buffer storage and are positioned between the A and B register classes.

$$RN_X = KN + (N_X - 1) \tag{1}$$

$$RN_E = 2 \times N_X \tag{2}$$

$$EN_Y = N_Y - 1 \tag{3}$$

Due to the requirement of 26 feature map data outputs as inputs to the PE array in this design, it can be configured as $N_X$ = 13 and $N_Y$ = 2. The computation order of the output feature maps is shown in Figure 12. The new computation order also affects the storage order of the original feature maps. The purpose of studying the new storage method is to store the feature map data as continuously as possible on-chip, facilitating reading and computation. Due to the limited on-chip memory space, the storage method for each layer is influenced by factors such as the size of the input feature maps and the number of channels. Taking an input feature map of size 104 × 104 × 32 as an example, where 104 represents the width and height of the input feature map, and 32 represents the number of channels. Due to the on-chip memory space allocated for storing feature maps in this design being 26 Kbyte, with each data size being 16 bits, it can be converted to a data volume size of $13 \times 2^{10}$.

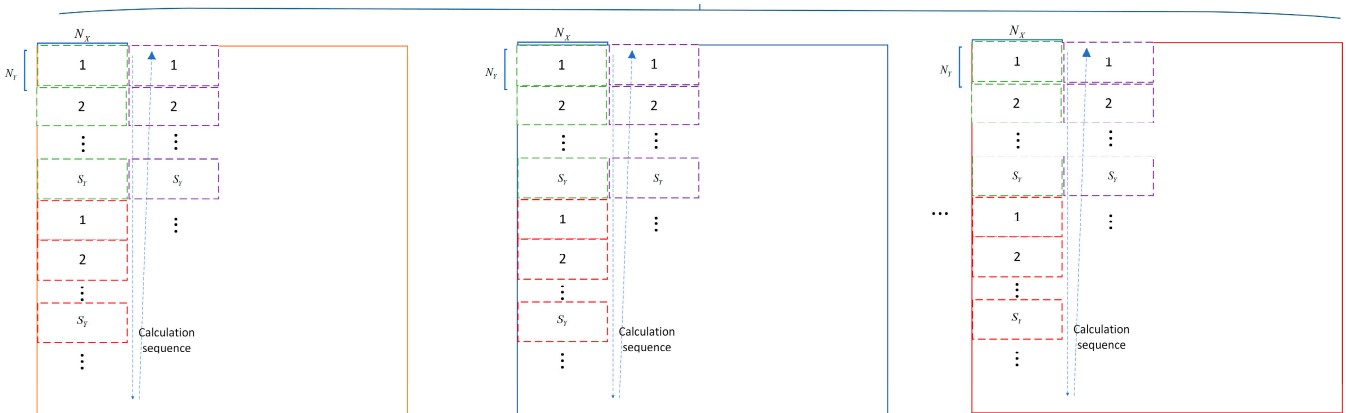

**Figure 12.** The calculation order of this design output feature map. Dashed boxes of different colors represent different output data tiles. The numbers within the dashed boxes represent the sequence of output data tiles, which are output in order. Dots indicate omissions. 32 represents the number of channels for parallel output.

By utilizing Equation (4), where $S_Y S_Y$ represents the maximum number of data blocks that can be stored along the Y direction on-chip, and $S_Y$ must be an integer, *NC* represents the number of channels in the input feature map, and TOTAL represents the total amount of data that can be stored on-chip, we can calculate that $S_Y$ equals 12, and each data block has a size of 15 × 4 × 32. Here, 15 is calculated from $N_X - 1 + KN$, and 4 is obtained from $N_Y - 1 + KN$. The storage process described above is shown in Figure 13.

$$[(N_X - 1) + KN] \times \{[KN + (N_Y - 1)] + 2 \times (S_Y - 1)\} \times NC \leq TOTAL \tag{4}$$

The circuit shown in Figure 11 corresponds to the FM_Reuse (FM stands for feature map) module in Figure 14. FM_control is a control module used to control the data flow within the FM_Reuse module and output the correct address signals to the FM BUFFER. This enables the efficient transfer of on-chip stored feature map data to the FM Reuse module. The Center_control module is responsible for connecting the FM_control and

WEIGHT_control modules, enabling them to exchange control signals with each other. This ensures that the feature map and weight data can be input to the PE array in aligned order.

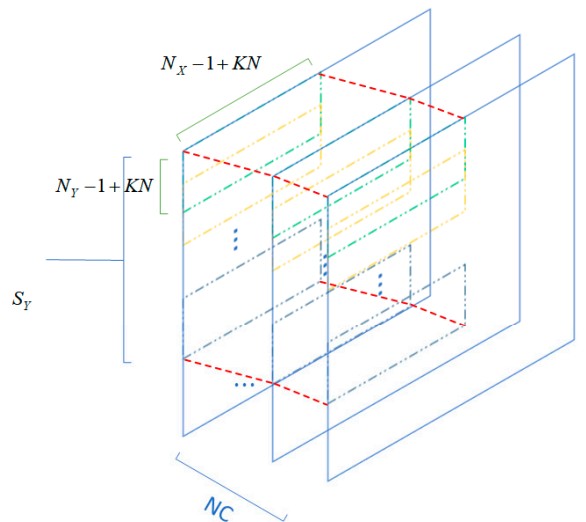
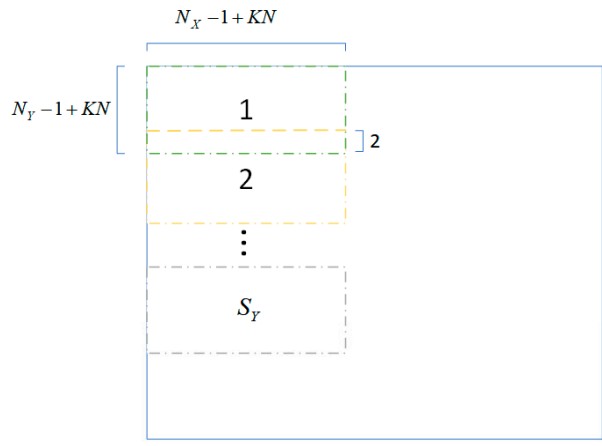

Three-dimensional view          Two-dimensional view

**Figure 13.** New input feature map storage order. Dashed boxes of different colors represent different input data tiles. The numbers within the dashed boxes represent the sequence of input data tiles, which are input in order. Dots indicate omissions.

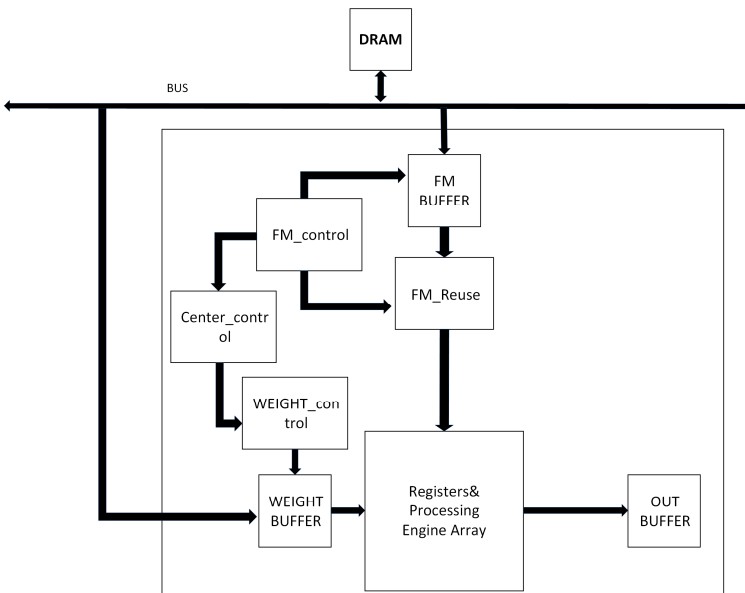

**Figure 14.** Structural diagram of accelerator section.

*5.2. Weight Data Reuse Module*

The method of weight data reuse is significantly different from the method of feature map reuse. The purpose of our design is to enable 32 different convolution kernels to be simultaneously input to the PE array for parallel computation. If the on-chip weight cache capacity is unable to store data for all 32 convolution kernels simultaneously, the number of times the convolution kernel data needs to be read from off-chip memory increases significantly. This number is proportional to the number of times the feature map data are read. Considering the layer with the maximum number of channels in the

convolutional kernels in the neural network algorithm, the amount of data that the on-chip weight cache should be able to accommodate is shown in Equation (5). $NW$ represents the maximum number of convolution kernel weights that can be stored on-chip, $NP$ represents the number of convolution kernels that are input in parallel, and $NC_{\max}$ represents the maximum number of channels in the convolution kernel.

$$NW = NC_{\max} \times NP \times KN \times KN \tag{5}$$

By using Equation (5) to calculate the maximum number of convolution kernels that can be stored in on-chip memory, it can be determined that the on-chip memory space for storing convolution kernels is 288 Kbytes after conversion. During the computation of an input feature map, the same convolution kernel needs to be reused multiple times. Since at least 32 complete convolution kernels are already stored on-chip, after completing one convolution operation, the read data address only needs to be adjusted to the beginning (for cases with only 32 convolution kernels) or jumped to the next set of 32 convolution kernels (for cases with more than 32 convolution kernels). For cases where the number of convolutional kernels is less than 32, the process is similar to when the number of kernels is exactly 32. After completing all convolution operations, it directly returns to the initial position. The above-mentioned storage and retrieval method of convolutional kernels is illustrated in Figure 15. This allows for the continuation of a new round of convolution operations. The control of the read address is performed by the WEIGHT_control module.

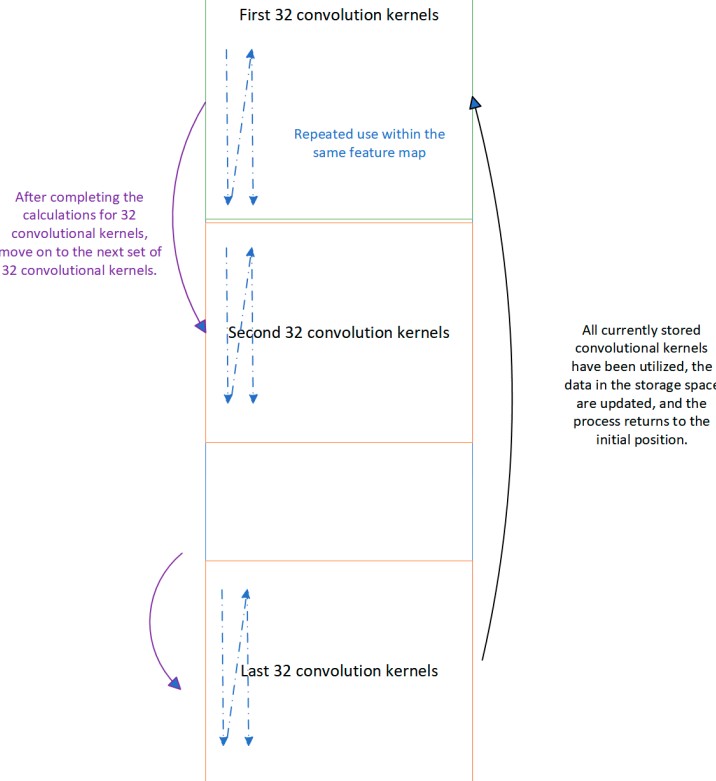

Storage and retrieval method of
convolutional kernels

**Figure 15.** Storage and retrieval method of convolutional kernels. Arrows represent the switching of data blocks after the switch condition is met.

## 6. Experiment Result and Analysis

### 6.1. Verification of the Effectiveness of the Data Reordering Module

As shown in Figure 16, from the red arrow to the solid yellow line in the figure, a total of nine clock cycles have passed (nine yellow arrows). The weight_re_data and a7_q serve as the output ports for weight data and one feature map tile datum, respectively. Assuming the initial nine feature map data range from decimal 1 to 9, and the weight data also range from decimal 1 to 9. The output data sequence of theoretical analysis is as follows: the weight data output is a 512-bit width ranging from 1 to 9, and one of the output ports (The total number is 26) of the feature map data, port a7_q, outputs nine numbers with a 16-bit width: 1, 0, 0, 0, 3, 0, 5, 0, 0. By combining the initially initialized original feature map and weight data in the design, it can be observed that the reordering module outputs correctly sorted data and completes data alignment. The above simulation results are achieved using Vivado's simulation tool.

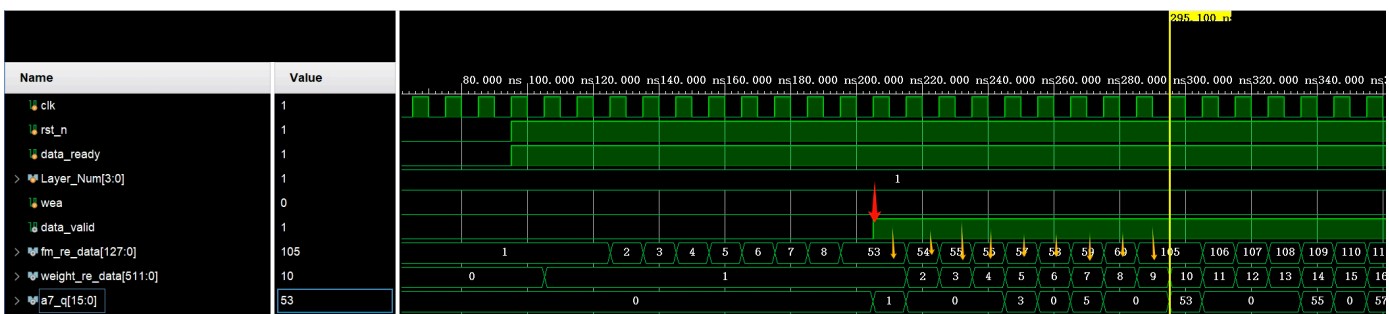

**Figure 16.** Simulation of the effect of data_reuse and weight_control modules. The red arrows represent the start of the valid timing. The yellow arrows represent the following nine valid clock cycles.

### 6.2. Analysis of the Reuse Effect of Input Feature Map Data

The accelerator with the added Data_reuse module is applied to the layers of the YOLOv4 Tiny, VGG16, and MobileNet v1 with a stride size of 1, and the number of on-chip and off-chip data exchanges for the input feature maps is obtained. The results are shown in Figure 17. Compared to the design without the added Data_reuse module, the number of data exchanges in each layer is significantly reduced. The reduction in the number of exchanges is particularly noticeable in the earlier layers with a larger amount of input feature map data. For YOLOv4 Tiny, the reduction rate of the maximum data exchange times is 82.9%. For VGG16, it is 84.92%. For MobileNet v1, it is 84.13%.

The inference time mainly consists of PE computation time and off-chip DRAM access time. The PE computation time equals the number of total multiply–accumulate (MAC) operations divided by the number of PE units and then multiplied by the time of one clock cycle. The DRAM access time equals the total amount of input data (stored in DRAM, including weights and feature maps) divided by the DRAM data bus width and then multiplied by the latency of a single DRAM access. The frequency of the accelerator is 100 MHz, and the latency of a single DRAM access is 37.5 ns [24]. Therefore, the inference time is 1.97 s without the on-chip data reordering module and 0.39 s with this module, which achieves 80.14% inference time reduction.

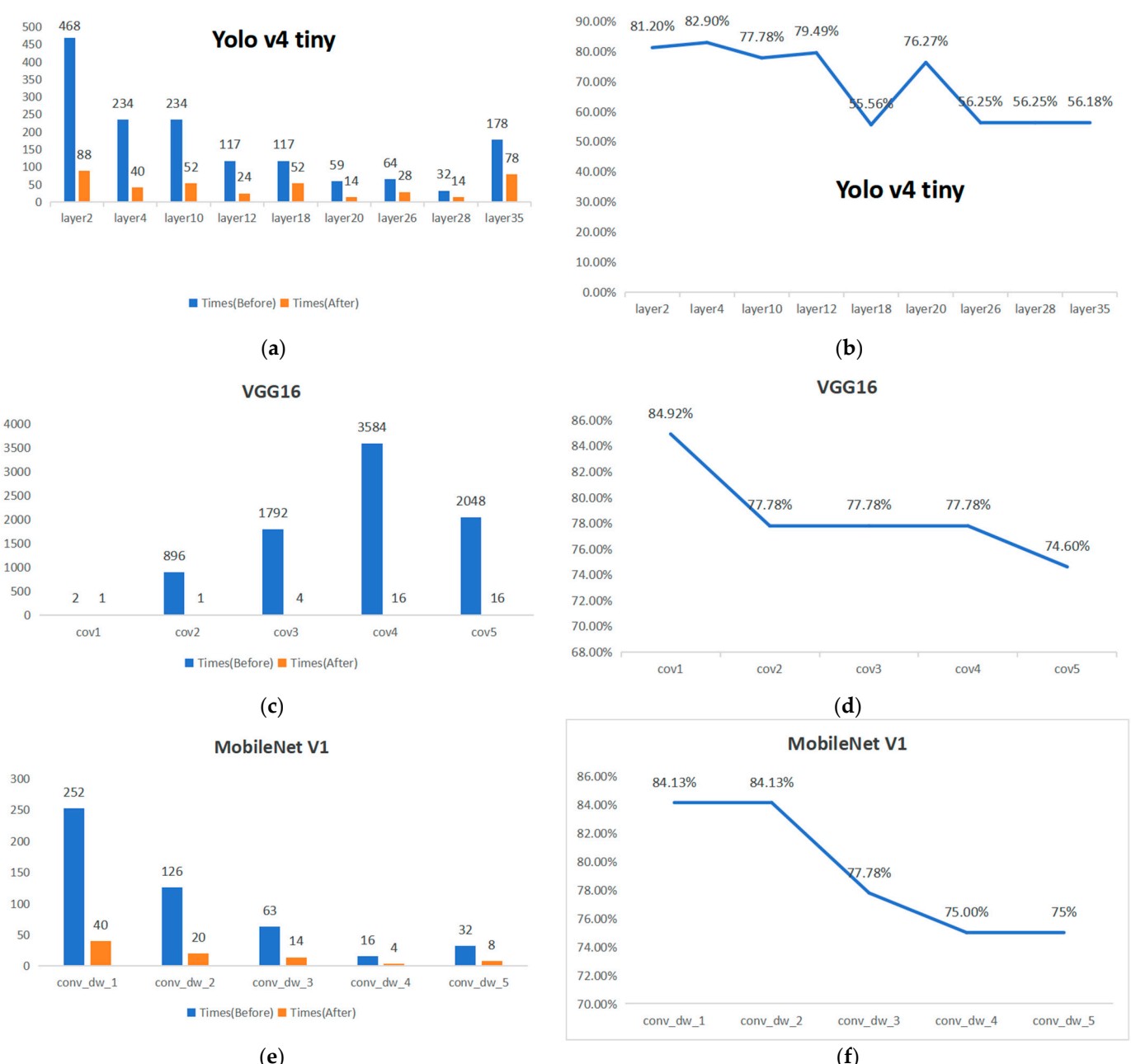

**Figure 17.** Comparison of on-chip FM BUFFER and off-chip data exchange times. (**a**,**c**,**e**) Comparison of the number of data exchanges before and after the application of the Data_reuse module at different layers. (**b**,**d**,**f**) Shows the variation curve of the reduction in the number of data exchanges at different layers.

### 6.3. Analysis of the Reuse Effect of Weight Data

With the inclusion of the Weight_control module, the weight data stored on-chip can be reused. Compared to the design without this module, the number of data exchanges between on-chip and off-chip is significantly reduced. The experimental results comparing YOLO v4 Tiny and VGG16 before and after the improvement are shown in Figure 18. They were obtained through calculations. The exchange frequency is greatly reduced, but this comes at the cost of increased on-chip storage space. The larger storage capacity allows for multiple accesses of the same set of weight data without the need for multiple transfers, thereby greatly reducing the number of data exchanges between on-chip and off-chip.

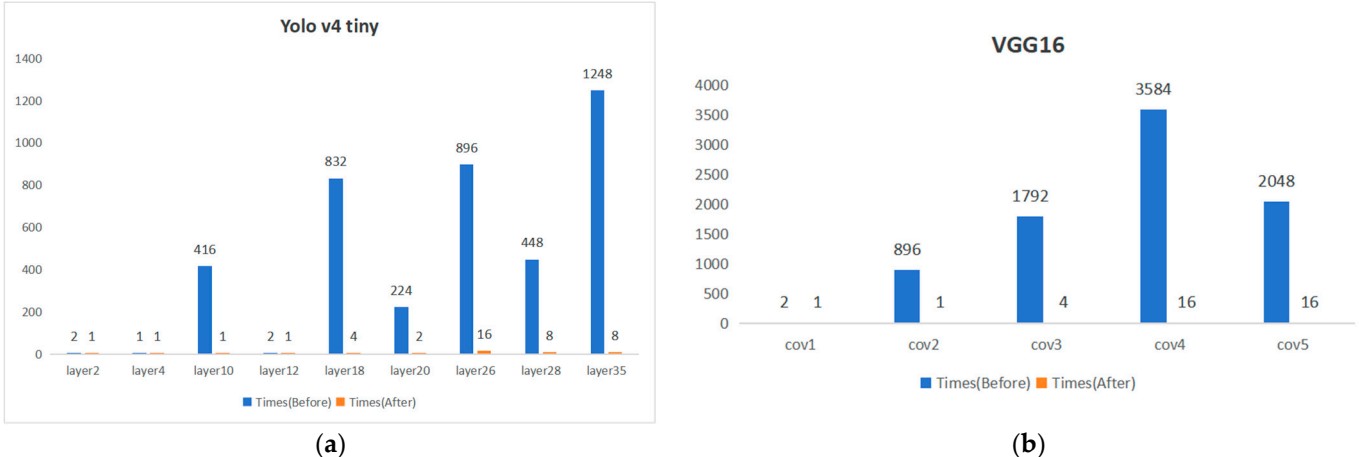

**Figure 18.** Comparison of the number of on-chip weight BUFFER and off-chip data exchanges after improvement (**a**) Yolo v4 tiny (**b**) VGG16.

### 6.4. Power Consumption Analysis

Table 2 presents a normalized comparison of the energy consumption for accessing different levels of storage structures and performing MAC operations. It is evident that the energy consumption for accessing external DRAM is significantly higher than the energy consumed for data movement in other storage forms. Optimizing this part of energy consumption is a key focus for accelerator energy optimization.

**Table 2.** Normalized energy cost relative to a MAC operation.

|  | DRAM | Global Buffer | Array | Register File |
|---|---|---|---|---|
| Norm. Energy | 200× | 6× | 2× | 1× |

Assuming that the power consumption for moving each datum from external DRAM to on-chip memory is equal, Table 3 presents the total data movement volume before and after the inclusion of the Data_reuse module. These data were obtained through statistical calculations. It can be observed that the data movement volume significantly decreases for different layers, with an average power reduction of 78.75%.

**Table 3.** Power consumption reduction rate of each layer.

| Layer Number/Average | Before/Number of Data | After/Number of Data | Reduce Rate |
|---|---|---|---|
| Layer 2 | 6,230,016 | 1,013,760 | 83.73% |
| Layer 4 | 3,115,008 | 499,200 | 83.97% |
| Layer 10 | 3,115,008 | 599,040 | 80.77% |
| Layer 12 | 1,557,504 | 276,480 | 82.25% |
| Layer 18 | 1,557,504 | 399,360 | 74.36% |
| Layer 20 | 785,408 | 161,280 | 79.47% |
| Layer 26 | 851,968 | 215,040 | 74.76% |
| Layer 28 | 425,984 | 107,520 | 74.76% |
| Layer 35 | 2,369,536 | 599,040 | 74.72% |
| Average |  |  | 78.75% |

### 7. Conclusions

This article presents the design and implementation of a configurable convolutional neural network (CNN) hardware accelerator. We analyze the key factors influencing the performance of the accelerator and propose an optimization scheme based on on-chip data reordering. Circuit modules were designed and implemented for sorting feature map and

weight data, which improved the utilization of on-chip data and reduced the number of data exchanges between on-chip memory and off-chip DRAM. The experimental results indicate that the power consumption is reduced on average by 78.75%. The number of external memory accesses is reduced by up to 82.9%, resulting in a maximum reduction of 82.9% in external memory access latency.

**Author Contributions:** Conceptualization, Y.Z. and X.H.; methodology, Y.L.; validation, M.N., M.C. and R.C.; writing—original draft preparation, Y.L.; writing—review and editing, X.H. and Y.L.; funding acquisition, L.C. All authors have read and agreed to the published version of the manuscript.

**Funding:** This research was funded by the National Key R&D Program of China under Grant 2022YFB4400400.

**Data Availability Statement:** The data presented in this study are available in this article.

**Conflicts of Interest:** The authors declare no conflicts of interest.

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
