# Peer review of "Design of a Convolutional Neural Network Accelerator Based on On-Chip Data Reordering"

_electronics, doi:10.3390/electronics13050975_

Round 1

Reviewer 1 Report

Comments and Suggestions for Authors

Please consider adding a figure showing the details of the proposed method. The new approach shown in Figure 12 is difficult to understand.

Please include the throughput results in the experiments.

Please show additional results using various CNN-based deep learning models, such as ResNet, DenseNet, MobileNet and EfficientNet.

Comments on the Quality of English Language

The paper is well-written.

Reviewer 2 Report

Comments and Suggestions for Authors

- It is difficult to determine exactly what the contribution is, so please revise the sentences to describe it accurately. Also, please describe how the results of the proposed technique improve upon existing research.

- Chapter or Section would be better than Part

   The remaining parts of the article are organized as follows. Part 2...

   -> The remaining chapters of the article are organized as follows. Chapter 2...

- Equation would be better than Formula

- All pictures must be modified to be consistent and readable.

- The caption of a table should be located above the table.

- As a result, it is necessary to compare the power of the entire HW when performing Yolov4, including the power consumption of the execution engine, rather than a reduction in the rate of data exchange or move rate. In addition, inference time must be presented and accuracy must also be compared. In addition, it is necessary to analyze the effectiveness of performance and energy savings compared to cost. Furthermore, it is necessary to present comparative results with other similar existing accelerator studies.

- In the Data Reuse Module, it is necessary to explain in detail how exactly data reordering is performed in the circuit (including the A, B, C, D, E register group and the exact operation between the registers for each group and detailed explanations in Figures 11 and 12).

- The Weight Data Reuse Module also requires a more accurate and detailed circuit operation explanation.

- It is necessary to mention the stability of operation even in cases where the number of convolution kernels is less than 32.

- Application results for models other than Yolov4 also need to be presented.

Comments on the Quality of English Language

English needs to be improved overall.

In particular, the spacing throughout the paper (including figure captions) is incorrect and needs to be corrected.

Reviewer 3 Report

Comments and Suggestions for Authors

This work meets the objective of presenting an optimization scheme based on on-chip data reordering for CNN acceleration. Congratulations to the authors. Below, I present some observations to make the work more meaningful:

Throughout the writing, you use "PE" a lot, but you never specify what it means. Please specify it the first time you use it, which I detect is on page 1, line 38.

Please specify what DCNN means, used on page 1, line 40.

On page 8, lines 199-213, you talk about two methods, but you don't use references. Where are these methods used? The way you explain it could be more explicit. Please make a comparative table of both methods to highlight the relevant aspects. Put your results in this table to denote your contribution.

On page 9, lines 219-220, you talk about a "significant increase". Please give quantitative approximations, as you did in previous examples, such as Figure 8. This way, the example will be more explicit.

On page 11, line 266, what do you mean by FM? Please explain.

On page 13, lines 303-308, you explain Figure 14. The figure itself is challenging to read. By increasing the size of the text and pointing out the points of interest in the image explained in this text, you will reach an impactful explanation of your results. For example, you did it correctly when referencing the yellow line. This way, you will be able to explain your results better.

Page 14, Table 1. Put "mac" in capital letters "MAC" and correct the word "relative".

Were the present results only simulated or also implemented in hardware? Please clarify it in the text. Please explain the simulation software and possible hardware used in the experiment.

On page 15, Table 2, you talk about power consumption. Was that consumption only calculated, or was it also measured? Please clarify it.

The main contribution of your article needs to be clarified. It is necessary to compare it with other methodologies. What is the scope of other methodologies? Is your methodology better or worse than other existing methodologies? Please reference other methodologies and compare them against your results.

Finally, I noticed that all your references are older than 2017, which means that they are not recent. It is imperative to have more recent references.

Round 2

Reviewer 2 Report

Comments and Suggestions for Authors

- As a reviewer, it can be seen throughout the paper that the paper has been revised and improved more than before.

- Redraw Figures 1 and 5 with more refinement.

  Also, redraw Figure 11 so that the contents are legible.

- At page 10, please clarify whether the second method is an existing technique or one proposed by the authors. If it is a proposal, please write it as "proposed method".

- Please compare and present the inference time according to the data exchange rate reduction and also compare the accuracy.

Comments on the Quality of English Language

There is a need to revise the English throughout the paper.

In particular, the spacing throughout the paper is incorrect and needs to be corrected.

Reviewer 3 Report

Comments and Suggestions for Authors

Thank you for answering all my questions and suggestions. 

Author Response

Thank you very much for your guidance on my article, and I appreciate your positive feedback.